# New Small-Molecule SERCA Inhibitors Enhance Treatment Efficacy in Lenvatinib-Resistant Papillary Thyroid Cancer

**DOI:** 10.3390/ijms251910646

**Published:** 2024-10-03

**Authors:** Jungmin Kim, Hang-Seok Chang, Hyeok Jun Yun, Ho-Jin Chang, Ki Cheong Park

**Affiliations:** 1Department of Surgery, Yonsei University College of Medicine, 50-1, Yonsei-ro, Seodaemun-gu, Seoul 03722, Republic of Korea; jm_kim@yuhs.ac; 2Department of Surgery, Thyroid Cancer Center, Gangnam Severance Hospital, Institute of Refractory Thyroid Cancer, Yonsei University College of Medicine, Seoul 06273, Republic of Korea; surghsc@yuhs.ac (H.-S.C.); gsyhj@yuhs.ac (H.J.Y.); docjang@yuhs.ac (H.-J.C.)

**Keywords:** papillary thyroid cancer, lenvatinib, drug-resistant, SERCA inhibitors

## Abstract

Papillary thyroid cancer (PTC) is one of the most treatable forms of cancer, with many cases being fully curable. However, resistance to anticancer drugs often leads to metastasis or recurrence, contributing to the failure of cancer therapy and, ultimately, patient mortality. The mechanisms underlying molecular differences in patients with metastatic or recurrent PTC, particularly those resistant to anticancer drugs through epigenetic reprogramming, remain poorly understood. Consequently, refractory PTC presents a critical challenge, and effective therapeutic strategies are urgently needed. Therefore, this study aimed to identify small-molecule inhibitors to enhance treatment efficacy in lenvatinib-resistant PTC. We observed an increase in sarco/endoplasmic reticulum calcium ATPase (SERCA) levels in patient-derived lenvatinib-resistant PTC cells compared with lenvatinib-sensitive ones, highlighting its potential as a therapeutic target. We subsequently identified two SERCA inhibitors [candidates 40 (isoflurane) and 42 (ethacrynic acid)] through in silico screening. These candidates demonstrated significant tumor shrinkage in a xenograft tumor model and reduced cell viability in patient-derived lenvatinib-resistant PTC cells when used in combination with lenvatinib. Our findings have potential clinical value for the development of new combination therapies to effectively target highly malignant, anticancer drug-resistant cancers.

## 1. Introduction

Thyroid cancer (TC) is the most common endocrine malignancy, characterized by the development of cancer cells in the thyroid gland in the neck [1,2]. TC is typically classified into four subtypes: medullary TC (MTC), follicular TC (FTC), papillary TC (PTC), and anaplastic TC (ATC) [3]. Clinically, TCs are categorized as either undifferentiated or differentiated [4,5]. Well-differentiated TC generally has a favorable prognosis, whereas poorly differentiated or undifferentiated TC (PDTC and UTC) is rare and associated with a poor prognosis [6,7]. Refractory TC, such as PDTC and UTC, exhibits resistance to anticancer drugs due to epigenetic reprogramming, leading to recurrence or metastasis and ultimately patient death [8]. Despite clear clinical behaviors, the biological and molecular mechanisms underlying drug sensitivity and resistance in these cancers require further investigation [9,10]. Consequently, research has focused on elucidating the differences between drug-sensitive and drug-resistant cancers, particularly through the analysis of mutations [11,12]. PTC is well known general endocrine malignancy, which represents almosty over 80% of whole well-differentiated TC, a 10 year survival of about 90%, it is regarded a slothful cancer [13]. Even though the properties of PTCs are well differentiated with a low rate of recurrences or metastases, partial drug resistant subclone shows aggressive variants, with distinct clinical and pathological features. Among mostly aggressive variants of PTC are the tall cell variant (TCV), diffuse sclerosing variant (DSV), solid variant (SV), hobnail variant (HV) and columnar cell variant (CCV). These variants have been involved with higher rates of recurrence and metastasis, as well as in several cases may have lower survival rate [14]. However, the molecular mechanisms of drug resistance in PTC remain poorly understood. This resistance, driven by epigenetic reprogramming, contributes significantly to the increased mortality associated with anticancer drug-resistant PTC, underscoring the need for effective therapeutic strategies [15]. Lenvatinib targets fibroblast growth factor receptors 1–4 (FGFR-1–4), vascular endothelial growth factor receptors 1–3 (VEGFR1–3), and platelet-derived growth factor receptor α (PDGFRα) in PTC. In lenvatinib-resistant cancer cells, several methods to aquire mechanism of lenvatinib-resistant were included epithelial-mesenchymal transition (EMT), RNA modification, translational modification and lenvatinib self target signal pathway [16].

This study aimed to identify small-molecule inhibitors to enhance treatment efficacy in lenvatinib-resistant PTC. The latent role of SERCA in cancer progression and survival has been a dynamic area of study given its role in cytosolic free calcium homeostasis and its influence on cell survival and ER stress pathway. Overall, these research propose that SERCA is a common mechanism to avoid apoptosis under acute ER stress condition [17]. We validated and identified inhibitors of sarco/endoplasmic reticulum calcium ATPase (SERCA), a crucial regulator of cytoplasmic free calcium levels [17,18,19], which are notably elevated in patient-derived lenvatinib-resistant PTC compared with lenvatinib-sensitive PTC. Cytoplasmic free calcium is implicated in various cellular processes, including those involved in cell survival and death, such as autophagy and apoptosis, particularly under severe endoplasmic reticulum (ER) stress conditions [19,20]. Our findings indicated that the newly identified SERCA inhibitors (compounds 40 and 42) are promising therapeutic options against refractory PTC, including lenvatinib-resistant PTC.

## 2. Results

### 2.1. Patient-Derived PTC Cell Lines and Their Properties

Three classes of PTC cell lines—YUMC-S-P2 (a patient-derived lenvatinib-sensitive PTC cell line), and YUMC-R-P7 and YUMC-R-P8 (lenvatinib-resistant PTC cell lines)—used in this study were derived from resected specimens of patients treated at Severance Hospital, Yonsei University College of Medicine, Seoul, Republic of Korea (Table 1). These patient-derived PTC cells were characterized cancer stemness gene expression profiling based on RNA-Seq analysis (Figure 1) and immunoblot assay (Appendix A). Immunoblot carried out antibody of B-Raf (V600E mutant specific and ELF3 [E26 transformation (ETS)-specific related transcription factor-3 (ELF3)] in BRAF wild-type TC (FTC133, CAL62 and ML1) and BRAF-mutant PTC (8505C,YUMC-S-P2, -R-P7 and –R-P8). BRAF mutation is associated with overexpression of ELF3 in PTC. PTC patients as lenvatinib-sensitive or -resistant was classified underwent lenvatinib after which the disease progression was confirmed in the lenvatinib response evaluation. In lenvatinib-resistant PTC patient, cancer recurrence and metastasis were caused after lenvatinib prescribed. The lenvatinib-resistant cell lines, YUMC-R-P7 and YUMC-R-P8, exhibited greater resistance than the lenvatinib-sensitive YUMC-S-P2. This resistance was associated with metastasis or recurrence, as documented in the pathology reports of these patients (Table 1).

### 2.2. Distinctions in Genetic Alterations and Activated Signaling Pathways between Patient-Derived Lenvatinib-Sensitive and -Resistant PTC Cell Lines

Cancer stem cell (CSC) properties and the activation of survival signaling pathways are more pronounced in certain cells that survive under severe ER stress conditions than in non-CSC cells [21,22,23,24,25]. These features are crucial in understanding how CSCs exhibit resistance to therapeutic agents [26,27]. Drug-resistant cancer cells often display traits associated with cancer stemness, as evidenced by several studies [20,22,24]. In this study, we identified genetic alterations through epigenetic reprogramming between drug-sensitive and -resistant cancer cells. To elucidate the genetic changes and the stimulated signaling pathways between lenvatinib-sensitive (YUMC-S-P2) and -resistant (YUMC-R-P7 and YUMC-R-P8) PTC, we conducted RNA sequencing (RNA-Seq) to perform a transcriptome analysis (Figure 1A–C). Based on RNA-Seq analysis, gene expression profiling revealed that, compared with YUMC-S-P2, YUMC-R-P7 and YUMC-R-P8 cells showed a significant upregulation of cancer stemness markers (*KRT17^high^*, *ALDH1A1^high^*, *CD133^high^*, *CD44^high^*, *SOX2^high^*, *KRT19^low^*, and *CD24^low^*) (Figure 1B, top). EMT is known as involved in drug resistance characteristic of cancer cells. Furthermore, FGF/FGFR signaling pathway involved in cancer development and drug resistance. In lenvatinib-resistant PTC, notable differences were also observed in cancer fibroblast growth factors and their receptors (*FGF1*, *FGF5*, *FGF11*, *FGF13*, *FGF16*, *FGFR2*, *FGFR3*, and *FGFR4*), and additional EMT markers [*zinc finger protein SNAIL1 (SNAIL1), SNAIL2, zinc finger E-box-binding homeobox 1 (ZEB1), ZEB2, and twist family bHLH transcription factor 1 (TWIST1)*] (Figure 1B, middle and bottom).

Particularly, KEGG pathway analysis indicated that signaling pathways related to calcium and cancer stemness, including Notch, Wnt, PPAR, PI3K/Akt, and TGF/SMAD, were significantly more activated in lenvatinib-resistant PTC than in lenvatinib-sensitive PTC (Figure 1C, top and bottom) [23,24,28,29,30]. We hypothesize that these highly activated calcium-related genes and signaling pathways in lenvatinib-resistant PTC cells are crucial in enabling PTC cells to evade cytoplasmic calcium-mediated apoptosis under severe ER stress conditions induced by drug treatment such as lenvatinib [23,28]. ER stress enhances the release of cytosolic free calcium from the ER to the cytosol via IP3R (inositol 1,4,5-trisphosphate) receptors, which is regulated by calcium pumps, exchangers, and channels to maintain cellular calcium homeostasis. However, inordinate elevation of cytosolic free calcium beyond physiological levels triggers apoptotic signals under acute ER stress conditions by drug treatment. SERCA is a pivotal regulator and therapeutic target in the regulation of cytosolic overburdened calcium in cancer [28,29].

The RNA-Seq analysis also highlighted a significant difference in the expression levels of SERCA (*ATP2A*) isoforms, known to be key regulators of calcium homeostasis. The basal levels of selective SERCA 1 expression among SERCA 2 and 3, which influence survival of drug-resistant PTC under severe ER stress conditions [18,31,32,33], were higher in YUMC-R-P7 and YUMC-R-P8 compared with those in YUMC-S-P2 (Figure 1D). Therefore, the use of lenvatinib-resistant PTC in the current study could be instrumental in developing therapeutic strategies for managing cancer metastasis or recurrence in patients with refractory PTC subtypes. The current results showes that the increase of SERCA expression could be therapeutic target in lenvatinib-resistant PTC cells.

### 2.3. Identification of Therapeutic Molecules, Candidates 40 and 42, Based on SERCA Structure through In-Silico Screening for Suppression of Lenvatinib-Resistant PTC

To provide a foundation for our findings, we hypothesized that the functional inhibition of SERCA could offer a viable clinical strategy for suppressing lenvatinib-resistant PTC cells. We screened numerous chemical compounds for their ability to bind with SERCA, assessing their pharmacophoric binding interactions via in-silico screening. The potential SERCA inhibitors were screened from virtual chemical library based on chemical binding similarity to the previously-known SERCA1 inhibitors. The chemical compounds that have high similarity to the known inhibitors (cutoff 0.75) were selected as a candidate for experimental validation. The number of selected molecules were six from synthetic chemical library and nine from approved or experimental drugs in DrugBank database. As expected, three drugs known to target SERCA1 (DrugBank ID DB04638, DB07604 and DB03909) showed very high SERCA-binding score (>0.8). Most of the candidates found in DrugBank are known to bind P-type ATPase family such as sodium/potassium-transporting ATPase subunit alpha-1 (AT1A1_HUMAN), potassium-transporting ATPase alpha chain 1 (ATP4A_HUMAN), and mitochondrial ATP synthase subunit delta (ATPD_HUMAN), which belong to the same protein family with SERCA1 (calcium-translocating P-type ATPase and HAD-IC family P-type ATPase). Notably, candidates 40 and 42 were identified and selected owing to their relatively high binding affinity to the molecular structure of SERCA. These candidates showed considerable suppression of SERCA function, leading to their selection as SERCA inhibitors in this study (Figure 2A). Candidates 40 and 42 represent therapeutic small molecules aimed at suppressing lenvatinib-resistant PTC. However candidate 40 and 42 alone treatment respectively was no considerably influenced to normal parathyroid cell in a dose-dependent manner (Appendix A). Lenvatinib treatment alone was showed siginificantly suppressed to cell viability of normal parathyroid cell (Appendix A).

We conducted cell viability and immunoblot analyses to evaluate the anticancer effects of these candidates on lenvatinib-sensitive (YUMC-S-P2) and lenvatinib-resistant (YUMC-R-P7 and YUMC-R-P8) (Figure 2B,C) PTC cells, both with lenvatinib alone and in combination with the SERCA inhibitors, candidates 40 or 42. The viability of lenvatinib-sensitive PTC cells (YUMC-S-P2) was significantly reduced in a dose-dependent manner following treatment with lenvatinib, regardless of the presence or absence of SERCA inhibitors (Figure 2B, left). In contrast, lenvatinib had no significant impact on the viability of lenvatinib-resistant PTC cells (YUMC-R-P7 and YUMC-R-P8) under the same conditions. However, the combined treatment of lenvatinib and SERCA inhibitors (thapsigargin as a positive control, along with candidates 40 and 42) significantly decreased the viability of lenvatinib-resistant PTC cells in a dose-dependent manner (Figure 2B, middle and right). Treatment with SERCA inhibitors alone did not significantly affect the viability of either lenvatinib-sensitive or -resistant PTC cells. The half-maximal inhibitory concentration (IC_50_) of the lenvatinib treatment alone was 12 µM in lenvatinib-sensitive PTC cells (Table 2). There is no considerable difference in the IC_50_ of lenvatinb alone or combined with SERCA inhibitors during treatment. Meanwhile, the anti-cancer influence of lenvatinib treatment alone showed no meaningful point in lenvatinib-resistant PTC. However, the anti-cancer influence of lenvatinib was significantly strong, when combined with SERCA inhibitors. The IC_50_ of combination with lenvatinib and SERCA inhibitors was respectively 12–25 µM in YUMC-R-P7 and -P8, lenvatinib-sensitive PTC cells (Table 2). Unlike lenvatinib-sensitive PTC, lenvatinib-resistant PTC cells showed a marked increase in the expression of BCL-2 and SERCA1 among the SERCA isoforms when treated with lenvatinib (Figure 2C). However, combination therapy with lenvatinib and the SERCA inhibitors (candidates 40 and 42) significantly increased markers of ER stress (CHOP) and apoptosis (cleaved-caspase 3) through the functional inhibition of SERCA (Figure 2C). Therefore, SERCA inhibitors may play a critical role in enhancing survival by managing the overload of cytoplasmic free calcium under severe ER stress conditions induced by lenvatinib in lenvatinib-resistant PTC.

### 2.4. SERCA1 as a Key Player in Lenvatinib-Resistant PTC Cells for Prolonging Survival under Lenvatinib Treatment

Previous research highlights SERCA as a crucial regulator of cytoplasmic free calcium-mediated apoptosis under severe ER stress conditions, particularly during anticancer drug treatment [18,31,33,34]. We observed a significant increase in SERCA1, among other SERCA isoforms, in lenvatinib-resistant PTC cells (YUMC-R-P7 and YUMC-R-P8) under severe ER stress induced by lenvatinib treatment than in lenvatinib-sensitive PTC (YUMC-S-P2) cells (Figure 3A). We conducted a cell viability assay using a calcium channel blocker (bepridil, verapamil, or nifedipine), NCX (Na^+^/Ca^2+^ exchanger) inhibitor (KB-R7943), plasma membrane calcium ATPase (PMCA) inhibitor (caloxin 2A1), and SERCA inhibitors (thapsigargin, C40, and C42), alone or in combination with lenvatinib. These tests demonstrated that, unlike NCX or calcium ion channels, SERCA played a pivotal role in prolonging survival in lenvatinib-resistant PTC cells under severe ER stress conditions (Figure 3B,C). When tested individually, none of the inhibitors significantly affected the viability of lenvatinib-resistant PTC cells (YUMC-R-P7 and YUMC-R-P8). Additionally, SERCA1 expression showed a considerable increase under lenvatinib treatment compared to conditions without lenvatinib.

Interestingly, despite this increase in SERCA1 expression in lenvatinib-resistant PTC cells, the ER stress marker CHOP was notably higher in the thapsigargin-treated group (a SERCA inhibitor) compared with groups treated with inhibitors of calcium channels, NCX, or PMCA (Figure 3D). Lenvatinib-resistant PTC was significantly increased SERCA1 expression but nevertheless only SERCA inhibitors treatment with lenvatinib group was showed considerably increase of CHOP, ER stress marker. Bepridil, verapamil, nifedipine, KB-R7943 and caloxin 2A1 were no significantly influenced to ER combination with lenvatinib.

These results demonstrated that in lenvatinib-resistant PTC cells, SERCA isoforms is the reasonable target for drug-resistnat PTC cells evading the cytoplasmic free calcium-mediated apoptosis under severe ER stress conditions produced by drug treatment such as lenvatinib.

### 2.5. Targeted Therapy In Vivo Treatment with Candidates 40 and 42 in a Patient-Derived Lenvatinib-Resistant PTC Cell Mouse Xenograft Model

To evaluate the anticancer effects of combining lenvatinib with candidates 40 and 42, we utilized a mouse xenograft model with both lenvatinib-sensitive (YUMC-S-P2) and -resistant (YUMC-R-P7 and YUMC-R-P8) PTC cells. Mouse xenograft model of lenvatinib-sensitive and -resistant cells were treated lenvatinib alone or in combination with SERCA inhibitors). In the lenvatinib-sensitive PTC cell xenograft model, tumor shrinkage was significantly induced by lenvatinib treatment, regardless combined with SERCA inhibitors (Figure 4A, top). However, lenvatinib treatment alone did not produce a considerable change in tumor volume in the lenvatinib-resistant PTC xenograft model (Figure 4B,C, top) In these lenvatinib-resistant PTC xenograft models, combination treatment with lenvatinib and SERCA inhibitors (thapsigargin as a positive control, candidates 40 and 42 as inhibitors) markedly increased tumor shrinkage (Figure 4B,C, top). The resected tumor weight corresponded closely with the changes in tumor volume (Figure 4A–C, middle). The treatment with all agents alone did not impact the overall body weight of the mice (Figure 4A–C, bottom). We conducted an immunoblot assay on total tumor tissue lysates to evaluate the relationship between SERCA and CHOP (a marker of ER stress) protein expression under severe ER stress conditions induced by lenvatinib treatment. SERCA1 expression showed no significant change in lenvatinib-sensitive PTC when treated with lenvatinib alone or in combination with SERCA inhibitors (Figure 5A). CHOP expression was increased in the lenvatinib-treated group regardless of the presence of SERCA inhibitors. In contrast, in lenvatinib-resistant PTC, SERCA1 expression was high under treatment with lenvatinib, with or without SERCA inhibitors (Figure 5B,C). Notably, despite the high increase of SERCA1 in lenvatinib-resistant PTC, CHOP levels did not increase, whereas ER stress was significantly elevated due to the functional inhibition of SERCA by the combination treatment with lenvatinib and the SERCA inhibitors identified in this study (Figure 5B,C).

Taken together these results propose that the SERCA inhibitors, candidate 40 and 42, would offer a new clinical approach to considerably induction of tumor shrink in refractory cancer such as drug-resistant PTCs.

## 3. Discussion

PTC is the most common type of thyroid cancer, accounting for approximately 80% of all thyroid carcinoma cases. It generally has a favorable prognosis and is curable. However, in rare instances, drug-resistant PTCs lead to recurrence or metastasis, resulting in a poor prognosis and potentially the death of the patient [35]. Notably, managing these refractory PTCs with existing clinical approaches is challenging [36,37]. Numerous cytogenetic events and oncogenic mechanisms contribute to the development of advanced TCs [38,39]. Further, the role of epigenetic reprogramming in the aggressiveness of refractory PTC remains unclear [39].

Previous research has demonstrated that preoperative chemotherapy can improve survival rates post-surgery, and many studies have confirmed the efficacy of combined chemotherapy and surgery, even when expedited treatment was previously deemed impractical [40,41]. However, effective therapies for neoadjuvant or basal-adjuvant treatment of drug-resistant cancers remain lacking [42,43], contributing to a significant number of patient deaths. Therefore, rational and reliable therapies are urgently needed for patients with drug-resistant cancer.

In this study, we identified new small molecules that may suppress lenvatinib-resistant and potentially other aggressive cancers through RNA-Seq analysis of patient-derived lenvatinib-sensitive and lenvatinib-resistant PTC cells. We particularly focused on the Notch and calcium signaling pathways among the 15 significantly activated signaling pathways in the lenvatinib-resistant PTC cells compared to those in lenvatinib-sensitive cells. Previous studies have explored the interactions between Notch and calcium signaling pathways [44,45,46], and notably, Notch signaling is regulated by SERCA [47,48]. Our findings indicate that the regulation of SERCA expression in lenvatinib-resistant PTC cells is a critical factor in prolonging survival under conditions treated with lenvatinib. Further, SERCA isoforms, particularly SERCA1, are promising targets for countering the evasion of cytoplasmic free calcium-mediated apoptosis in lenvatinib-resistant PTC cells under severe ER stress conditions induced by anticancer drug treatment.

The study findings will be beneficial for devising future therapeutic approaches for refractory cancers. This study highlights therapeutic access based on epigenetic changes as a practical approach for the management of patients with refractory cancers. PTC is s generally known to be well-treated and high-survival rate. However unfortunately in not a few cases, exhibits resistance to anti-cancer drugs due to epigenetic reprogramming, leading to recurrence or metastasis and ultimately patient death. Drug resistance mainly derives from unusual, but highly probable presence of CSCs. Ability to differentiation and self-renewal into heterogeneous cancer cells, and hiding phenotypically and morphologically distinct cells are prominent prpperties of CSCs. They own some mechanisms that support them to survive even after disclosure to chemotherapy drugs by epigenetic reprogramming. Even though chemotherapy is able to destruct whole tumor cells, CSCs are left almost undamaged, and cause to drug-resistant. In current stduy was identificatied small molecules based on only one survival mechanism among a lot of survival related mechanisms in lenvatinib-resistant PTC compare then lenvatinib-sensitive PTC under acute ER stress conditions by lenvatinib treatment. In these lenvatinib resistant PTC was avoid to microenvironment with overload cytosolic free calcium-mediated apoptosis by increase of SERCA. Of course, the role of epigenetic reprogramming in the aggressiveness of refractory PTC remains unclear. We believe that our study makes a significant contribution to the literature because the scientific content of this manuscript successfully builds upon previous studies, and is reflective of a deep understanding regarding the molecular and genetic mechanisms underlying lenvatinib-resistant cancer. We believe that our study can significantly improve therapeutic approaches towards cancer, particularly those resistant to drugs. We prioritized the investigation of key genes and signaling pathways involved in managing excessive free calcium to enhance survival under severe ER stress conditions caused by lenvatinib treatment in lenvatinib-resistant PTC cells. The newly identified small molecules, candidates 40 and 42, significantly increase cell death in lenvatinib-resistant PTC cells through the functional inhibition of SERCA under severe ER stress conditions induced by lenvatinib treatment. Therefore, the SERCA inhibitors, candidates 40 and 42, could provide a new clinical strategy to significantly induce tumor shrinkage in refractory cancers, such as drug-resistant PTCs. However, several further studies were needed to the restrictions of only few patient results. To breakthrough these restrictions, not inconsiderable researchs are on going on diverse cases of patient-derived drug-resistant cancer.

## 4. Materials and Methods

### 4.1. Study Design and Ethical Considerations

Current research was showed retrospective study, which single central analysis of patients with PTC, detail information was indicated in our previous study [22,29,49]. All procedures involving patients were performed in accordance with the institutional ethical standards, all applicable local/national regulations, and guidelines of the 1964 Helsinki Declaration and its later amendments. In accordance with the Bioethics and Safety Act of Republic of Korea, formal written consent was not required for this type of retrospective, observational analysis. The study protocol was approved by the Institutional Review Board (IRB) of Severance Hospital, Yonsei University College of Medicine (IRB protocol: 3-2022-0331). Cell samples were isolated from patient specimen at the Severance Hospital, Yonsei University College of Medicine, Seoul, Republic of Korea.

### 4.2. Patients

#### 4.2.1. Patient 1

YUMC-S-P2 was 53-year-old man with papillary thyroid cancer. This patient had bilateral thyroid tumors with extrathyroidal extension. This patient underwent bilateral total thyroidectomy and bilateral modified radical neck dissection with central compartment neck dissection. Surgical findings showed that the tumor invaded the recurrent laryngeal nerve and was removed by careful shaving. After surgery, she was given 3 times high-dose radioiodine ablation therapy. Currently, radiologic examination and thyroid hormone tests are being followed without recurrence.

#### 4.2.2. Patients 2 and 3

YUMC-R-P7 (patient 2) and -P8 (patient 3) were 57 and 52-year-old woman and man with papillary thyroid cancer. This patient had multiple tumors and extensive extrathyroidal extension. This patient underwent bilateral total thyroidectomy with central compartment neck dissection. One year after surgery, metastasis to the mediastinum and right lateral cervical lymph nodes was confirmed, and she underwent mediastinal dissection through partial sternotomy and right modified radical neck dissection. The specimens for culture were obtained after the last operation. This patient underwent sorafenib after which the disease progression was confirmed in the sorafenib drug response evaluation. Currently, cancer recurrence and metastasis were caused and confirmed after lenvatinib prescribed.

### 4.3. Patient Tissue Specimens

Fresh tumor specimen was dissected from patients with biochemical and histologically proven PTC who were cured at the Severance Hospital, Yonsei University College of Medicine, Seoul, Republic of Korea. Fresh tumors were collected throughout surgical excision of PTC metastatic and primary sites.

### 4.4. Primary Culture and Cancer Cell Isolation

The patient-derived cancer cells were obtained from fresh tumors of patients. YUMC-S-P2, YUMC-R-P7 and -8 were obtained from papillary thyroid cancer patients treated at the Severance Hospital, Yonsei University College of Medicine, Seoul, Republic of Korea. After resection, tumors were kept in phosphate-buffered saline (PBS) with antifungal and antibiotics and moved to the laboratory. Normal tissue and fat were eliminated and rinsed with 1× Hank’s Balanced Salt Solution. Tumors were minced in a tube with dissociation medium containing DMEM/F12 with 20% fetal bovine serum supplemented with 1 mg/mL collagenase type IV (Sigma-Aldrich, St. Louis, MO, USA; C5138). The isolated cancer cells were grown in Dulbecco′s Modified Eagle′s Medium, supplemented with 10% fetal bovine serum. Candidates 40 (Sigma-Aldrich) and 42 (Sigma-Aldrich) treatment was diluted with PBS. Mycoplasmal contamination was checked for with the Lookout Mycoplasma PCR Detection Kit (Sigma-Aldrich, St. Louis, MO, USA; MP0035). Further protocol details are as described in our previous article [25].

### 4.5. mRNA-seq Data

We preprocessed the raw reads from the sequencer to remove low quality and adapter sequences before analysis and aligned the processed reads to the Homo sapiens genome assembly (GRCh37) using HISAT v2.1.0 (HISAT2, RRID: SCR 015530) [50]. HISAT utilizes two types of indexes for alignment: a global, whole-genome index, and tens of thousands of small local indexes. Both are constructed using the same Burrows–Wheeler transform (BWT) or graph FM index (GFM) as Bowtie2 (Bowtie 2, RRID: SCR 016368). Because of the use of these efficient data structures and algorithms, HISAT generates spliced alignments several times faster than Bowtie and the widely used BWA (BWA, RRID: SCR 010910). The reference genome sequence of Homo sapiens (GRCh37) and annotation data were downloaded from the National Center for Biotechnology Information (NCBI). Then, transcript assembly of known transcripts was processed using StringTie v2.1.3b (StringTie, RRID: SCR016323) [51,52]. Based on these results, expression abundance of transcript and gene were calculated as read count or fragments per kilobase of exon per million fragments mapped (FPKM) value per sample. The expression profiles were used for additional analyses, such as of differentially expressed genes (DEGs). In groups with different conditions, differentially expressed genes or transcripts were filtered through statistical hypothesis testing.

### 4.6. Statistical Analysis of Gene Expression Level

We performed statistical analyses to find differentially expressed genes using the estimates of abundances for each gene in the samples. Genes with one more than zeroed Read Count values in the samples were excluded. To facilitate log2 transformation, 1 was added to each Read Count value of filtered genes. Filtered data were log2-transformed and subjected to trimmed mean of M-values (TMM) normalization. The statistical significance of the differential expression data was determined using exactTest, edgeR and fold change, in which the null hypothesis was that no difference exists among groups. False discovery rate (FDR) was controlled by adjusting the *p*-value using the Benjamini-Hochberg algorithm. For DEG sets, hierarchical clustering analysis was performed using complete linkage and Euclidean distance as a measure of similarity. Gene-enrichment and functional annotation analysis and pathway analysis for significant gene list were performed based on Gene Ontology and KEGG pathway analyses.

### 4.7. Hierarchical Clustering

Hierarchical clustering analysis was carried out with complete linkage and Euclidean distance as calculate of resemblance to indicate the expression patterns of dissimilarly indicated transcripts which are satisfied with |fold change| ≥ 2 and independent *t*-test raw *p* < 0.05. All data analysis and visualization of dissimilarly indicated genes was directed with R 3.5.1 (www.r-project.org, accessed on 16 March 2022).

### 4.8. Total RNA Extraction and Quantitative Reverse Transcription-Polymerase Chain Reaction

Total RNA was extracted from tumor cells using the RNeasy Mini Kit (Qiagen, Germany, Cat# 74106) and One-Step reverse transcription-polymerase chain reaction (RT-PCR) Kit (Qiagen, Germany, Cat#204243) according to the manufacturer’s protocols. All data were normalized to *α-tubulin* expression. The following primers for *SERCA1*, *SERCA2*, and *SERCA3* were used for quantitative RT-PCR (qRT-PCR) analysis: *SERCA1*, 5′-GTGATCCGCCAGCTAATG-3′ (forward) and 5′-CGAATGTCAGGTCCGTCT-3′ (reverse); *SERCA2*, 5′-GGTGGTTCATTGCTGCTGAC-3′ (forward) and 5′-TTTCGGACAAGCTGTT GAGG-3′ (reverse); *SERCA3*, 5′-GATGGAGTGAACGACGCA-3′ (forward) and 5′-CCA GGTATCGGAAGAAGAG-3′ (reverse); and α-tubulin; 5′- CGGGCAGTGTTTGTAGACTTGG-3′ (forward) and 5′-CTCCTTGCCAATGGTGTAGTGC-3′ (reverse).

### 4.9. Cell Viability Assay

Cell viability was calculated by the MTT (3-(4,5-Dimethylthiazol-2-yl)-2,5-Diphenyltetrazolium Bromide) assay, cells were seeded in 96-well plates at 8 × 10^3^ cells per well and cultured overnight to achieve over 80% confluency. The detailed protocol can be found in [24]. Data were indicated as a percentage of the signal observed in vehicle-treated cells and are shown as the means ± SEM of triplicate experiments.

### 4.10. Immunoblot Analysis

The primary antibodies SERCA1 (1:500, Abcam, Cambridge, UK, Cat# 133275), SERCA2 (1:500, Abcam, Cat# 137020), SERCA3 (1:300, Abcam, Cat# 154259), C/-EBP homologous protein (CHOP, 1:100, Santa Cruz Biotechnology, Cat# 7351), Bcl-2 (1:500, Cell Signaling Technology, Beverly, MA, USA, Cat# 4223S), caspase-3 (1:500, Cat# 9661, Cell Signaling Technology) and β-actin (1:2000, Santa Cruz Biotechnology, Cat# 47778) were purchased and maintained overnight at 4 °C. The detailed protocol can be found in our previous article [22,24,29,49].

### 4.11. Human PTC Cell Xenograft

All experiments were approved by the Animal Experiment Committee of Yonsei University. YUMC-S-P2, YUMC-R-P7 and -P8 patient-derived PTC cells (6.2 × 10^6^ cells/mouse) were cultured in vitro and then injected subcutaneously into the upper left flank region of female NOD/Shi-scid, IL-2Rγ KOJic (NOG) mice. After 14 days, tumor-bearing mice were grouped randomly (n = 10 per group) and treated 25 mg/kg SERCA inhibitors, thapsigargin, candidate 40 and 42 (p.o.) with 10 mg/kg lenvatinib (p.o.) either alone or combination (excluded for combination of SERCA inhibitors). Tumor size was measured every two day using calipers. Tumor volume was gauged by following method: L × S^2^/2 (L, longest diameter; S, shortest diameter). Animals were maintained under specific pathogen-free conditions, and all experiments were approved by the Animal Experiment Committee of Yonsei University (IACUC approval No 2022-0105).

### 4.12. Statistical Analysis

Statistical analyses were performed using GraphPad Prism 6.0 software (GraphPad Software, La Jolla, CA, USA), Microsoft Excel (Microsoft Corp, Redmond, WA, USA), and R version 2.17. One-way ANOVA was performed for the multi group analysis, and two-tailed Student’s t-test was performed for the two-group analysis. Values were expressed as mean ± standard error of mean. *p* values < 0.05 were considered statistically significant.

### 4.13. Virtual Screening with Chemical Binding Similarity

The potential SERCA-binding chemical compounds were screened by evolutionary chemical binding similarity (ECBS), which is built based on classification similarity-learning to prioritize evolutionarily-related chemical pairs (ERCPs). By ECBS, chemical pairs are considered as “similar” when their binding targets are identical or evolutionarily related. Among variants of ECBS models, the target-specific ensemble ECBS (TS-ensECBS) model was adapted for the virtual screening owing to the highest test accuracy in our previous study1. TS-ensECBS model was built for SERCA1 (i.e., the ERCPs are only defined for SERCA1 and its homologous proteins). The model was then used to calculate chemical binding similarity (ECBS score) between previously-known seven SERCA1 inhibitors (obtained from DrugBank and BindingDB database) and the virtual chemical library (141,102 chemicals combined from Maybridge and Chembridge screening collection, and DrugBank). The maximum ECBS score assigned for each molecule in chemical library was considered as a final SERCA-binding score. Thus, the output similarity score ranges from 0 to 1, and the scores closer to 1 represent higher binding probability to SERCA1. More details about the ECBS model can be found in our previous work [53].

## 5. Conclusions

SERCA plays a crucial role in managing overloaded free calcium under severe ER stress conditions induced by anticancer drugs, such as lenvatinib. Our findings revealed that SERCA1, one of the SERCA isoforms, is predominantly expressed in patient-derived lenvatinib-resistant PTC cells under genotoxic stress caused by lenvatinib. Further, our results demonstrated significant tumor shrinkage in patient-derived lenvatinib-resistant PTC cells, both in vitro and in vivo, when treated with the SERCA inhibitors identified in this study. These findings support the efficacy of new combination therapies that incorporate these SERCA inhibitors to target highly refractory cancer cells, including those resistant to anticancer drugs.

Collectively, our study suggests that clinical approaches based on the genetic and signaling pathway differences between patient-derived lenvatinib-sensitive and lenvatinib-resistant PTC could be effective against refractory drug-resistant TCs. These findings could help establish prospective clinical strategies against refractory TC. However, further research is necessary to develop a targeted therapeutic approach for various drug-resistant cancer subtypes.

## Figures and Tables

**Figure 1 ijms-25-10646-f001:**
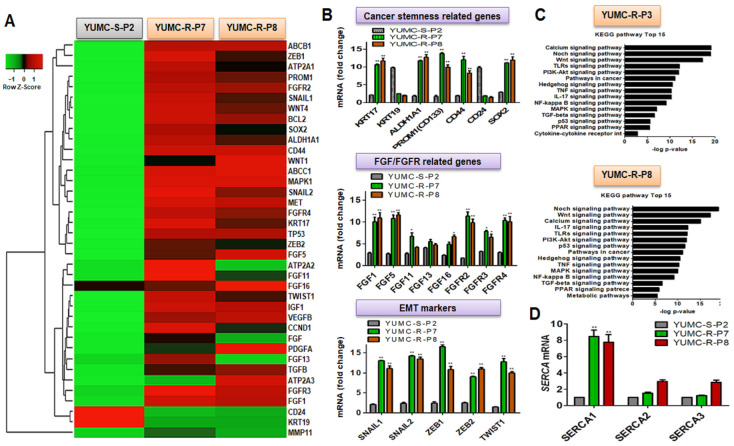
Characteristics of the patient-derived papillary thyroid cancer (PTC) cell lines used in this study. (**A**) Hierarchical clustering of gene expression differences between patient-derived lenvatinib-sensitive and -resistant PTC cells. (**B**) Analysis of gene expression transitions based on RNA-Seq data, focusing on cancer stem cell (CSC) markers, fibroblast growth factor (FGF) and FGF receptor-related genes, and epithelial-mesenchymal transition (EMT) markers in lenvatinib-sensitive and -resistant PTC cells. (**C**) Bar plot showing 15 significant pathways induced in the lenvatinib-resistant PTC cells, with comparisons between YUMC-R-P7 (top) and YUMC-R-P8 (bottom). (**D**) Variations in SERCA isoform-dependent RNA expression between lenvatinib-sensitive and lenvatinib-resistant PTC cells under basal conditions. * *p* < 0.05 vs. lenvatinib-sensitive PTC cells, YUMC-S-P2; ** *p* < 0.01 vs. lenvatinib-sensitive PTC cells, YUMC-S-P2.

**Figure 2 ijms-25-10646-f002:**
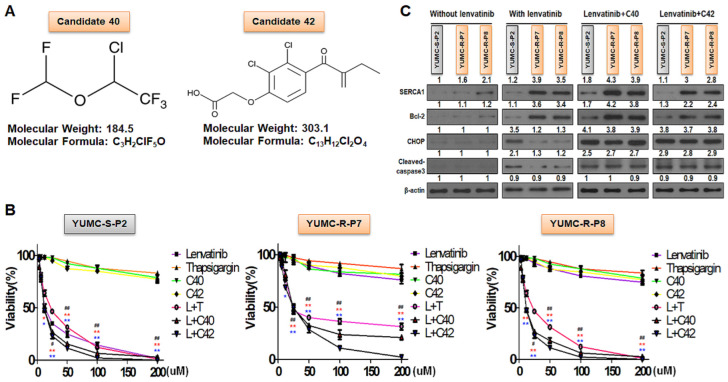
Identification and validation of SERCA inhibitors through virtual screening. (**A**) Chemical properties of the SERCA inhibitors, candidates 40 and 42. (**B**) Cell viability assay of lenvatinib-sensitive (left) and lenvatinib-resistant (middle: YUMC-R-P7, right: YUMC-R-P8) PTC cells exposed to the SERCA inhibitors (thapsigargin; positive control, candidates 40 and 42) in combination with lenvatinib. Points represent the mean percentage of values relative to the solvent-treated control. (**C**) Immunoblot analysis showing the effects of combining lenvatinib with the SERCA inhibitors on both lenvatinib-sensitive and lenvatinib-resistant PTC cells. All experiments were conducted in triplicate. Data are presented as mean ± standard deviation. * *p* < 0.05 and ** *p* < 0.01 versus control. L; Lenvatinib, T; Thapsigargin, C40; Candidates 40, C42; Candidates 42, ^#^,^##^; combination treatment with lenvatinib and thapsigagin, *,** (red); combination treatment with lenvatinib and C40, *,** (blue); combination treatment with lenvatinib and C42.

**Figure 3 ijms-25-10646-f003:**
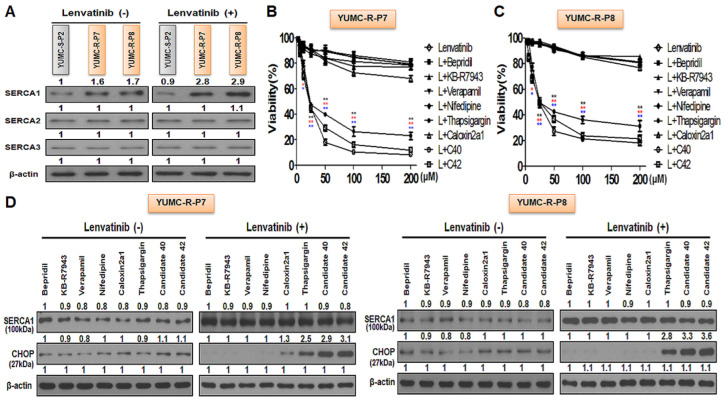
SERCA as a critical factor in enhancing survival under severe ER stress conditions induced by lenvatinib treatment. (**A**) Immunoblot analysis to assess changes in SERCA isoforms in patient-derived lenvatinib-sensitive and -resistant PTC cells, with and without lenvatinib treatment. (**B**,**C**) Cell viability assays using a calcium channel blocker (nifedipine), Na^+^/Ca^2+^ exchanger (NCX) inhibitor (KB-R7943), PMCA inhibitor (caloxin 2A1), and SERCA inhibitors (thapsigargin as a positive control, and candidates C40 and C42), both alone and in combination with lenvatinib. (**D**) Immunoblot analysis to detect changes in SERCA and CHOP (an ER stress marker) levels following treatment with a calcium channel blocker (nifedipine), NCX inhibitor (KB-R7943), PMCA inhibitor (caloxin 2A1), and SERCA inhibitors (thapsigargin as a positive control, and candidates C40 and C42), either alone or in combination with lenvatinib. * *p* < 0.05 and ** *p* < 0.01 versus control. L; Lenvatinib, T; Thapsigargin, C40; Candidates 40, C42; Candidates 42, ***,****; combination treatment with lenvatinib and thapsigagin, *,** (red); combination treatment with lenvatinib and C40, *,** (blue); combination treatment with lenvatinib and C42.

**Figure 4 ijms-25-10646-f004:**
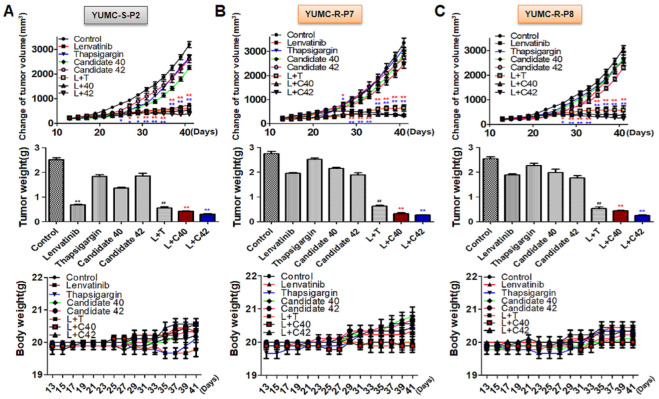
Enhanced tumor shrinkage in the xenograft model of patient-derived lenvatinib-resistant PTC cells following treatment with SERCA inhibitors combined with lenvatinib. (**A**–**C**, top) Changes in tumor volume over time. (**A**–**C**, middle) Weight of excised tumors. (**A**–**C** bottom) Changes in body weight of each group (n = 10). Tumor size was measured in NOD/Shi-scid IL-2Rγ KOJic (NOG) mice; the mice received treatments with lenvatinib combined with SERCA inhibitors, candidates 40 or 42, or each agent alone. Data are presented as mean ± standard error of the mean. * *p* < 0.05 and ** *p* < 0.01, compared with control. L; Lenvatinib, T; Thapsigargin, C40; Candidates 40, C42; Candidates 42, **; treatment with lenvatinib alone, **^##^**; combination treatment with lenvatinib and thapsigagin, *,** (red); combination treatment with lenvatinib and C40, *,** (blue); combination treatment with lenvatinib and C42.

**Figure 5 ijms-25-10646-f005:**
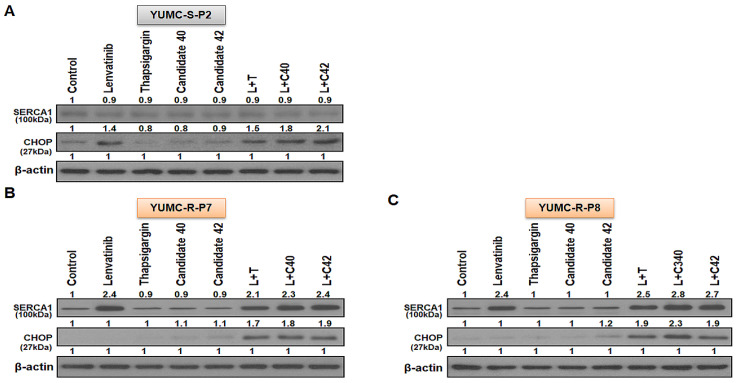
(**A**–**C**), Immunoblot analysis with SERCA1 and CHOP (endoplasmic reticulum stress marker) in a mouse xenograft model with lenvatinib-sensitive (YUMC-S-P2) and -resistant (YUMC-R-P7 and -P8) PTC cells. Beta-actin served as a loading control.

**Table 1 ijms-25-10646-t001:** Clinical characteristics of patients in the current study. Patient-derived papillary thyroid cancer (PTC) cells were isolated from specimens collected from these patients.

	YUMC-S-P2	YUMC-R-P7	YUMC-R-P8
Age at Diagnosis	53	57	52
Gender	Male	Female	Male
Primary Disease Site	Thyroid	Thyroid	Thyroid
Stage	T4aN1bM0	T4aN1bM1	T4aN1bM1
Primary Pathology	Papillary thyroid cancer	Papillary thyroid cancer (Recurrence & Metastasis after lenvatinib treatment)	Papillary thyroid cancer (Recurrence & Metastasis after lenvatinib treatment)
Classification of specimen used for culture	Fresh tumor	Fresh tumor	Fresh tumor
Obtained from	Severance Hospital, Seoul, Republic of Korea	Severance Hospital, Seoul, Republic of Korea	Severance Hospital, Seoul, Republic of Korea

**Table 2 ijms-25-10646-t002:** IC_50_ values for combinational value of SERCA inhibitors with lenvatinib in lenvatinib-sensitive and -resistant PTC cells. Each data point infers the mean of three particular MTT assays, carried out in triplicate. SEM, standard error of the mean; MTT, 3-(4,5-dimethylthiazol-2-yl)-2,5-diphenyltetrazolium bromide; IC_50_, half-maximal inhibitory concentration. L; lenvatinib, T; Thapsigargin, C40; Candidate 40, C42; Candidate 42.

Cell Line	Histopathology	Animal	Cell Proliferation IC_50_ (μM)			
			Lenvatinib	L + T	L + C40	S + C42
YUMC-S-P2	Thyroid, Papillary	Human	12 (±0.1)	12 (±0.2)	12 (±0.2)	12 (±0.2)
YUMC-R-P7	Thyroid, Papillary	Human	─	12 (±0.3)	25 (±0.2)	23 (±0.1)
YUMC-R-P8	Thyroid, Papillary	Human	─	23 (±0.2)	13 (±0.2)	13 (±0.4)

## Data Availability

The data from this study are available upon reasonable request from the corresponding author.

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
