# Peer review of "New Small-Molecule SERCA Inhibitors Enhance Treatment Efficacy in Lenvatinib-Resistant Papillary Thyroid Cancer"

_ijms, 2024, doi:10.3390/ijms251910646_

Round 1

Reviewer 1 Report

Comments and Suggestions for Authors

The authors attempted to show two novel SERCA inhibitors to cause death of PTC cells, especially effective against LEN-insensitive cells. Although the data appear interesting, there are a few important issues.

1) First of all, the authors only used in-silico assay of the two novel compounds. Assay should be performed to show they really inhibit SERCA using Ca2+ transport assay, eg. 45Ca2+ flux using microsomes, or fura-2 measurement.

2) Related to above, even if the compounds inhibit SERCA, validation of their selectivity has to be done,  ie, do they inhibit Na pump, PM Ca2+ pump, NCX, SOCC channels etc., using direct measurement. The findings that selective inhibitors did not potentiate LEN in killing resistant PTC cells did not add too much merit (Fig3BC) if the above direct assays are not performed.

3) Fig.2and 3 B,C extremely difficult to read because of the small sizes of symbols. Importantly, I suppose the x-axis is the concentration of the novel compounds, then how about the LEN alone group (LEN also 0 - 200 uM?)? Then, what was the LEN concentration when added together with novel compounds? What were the concentrations of other blockers or inhibitors, such as TSG and nifed etc?

4) I find the concentrations of drugs are missing in the figures or figure legends. This is an important issue.

5) Why are L-type channel blockers used? I wonder if PTC cells have L-type channels. Why not try SOCC or TRP blockers

6) Western blot results should be quantitated. 

Comments on the Quality of English Language

ok

Author Response

Reviewer 1

The authors attempted to show two novel SERCA inhibitors to cause death of PTC cells, especially effective against LEN-insensitive cells. Although the data appear interesting, there are a few important issues.

1) First of all, the authors only used in-silico assay of the two novel compounds. Assay should be performed to show they really inhibit SERCA using Ca2+ transport assay, eg. 45Ca2+ flux using microsomes, or fura-2 measurement.

 > Reply: I don't know how to thank you enough for reviewing our manuscript. A point-by-point response to your expert comments was indicated below. Thank you again for your review. I hope you are always healthy and happy!!

2) Related to above, even if the compounds inhibit SERCA, validation of their selectivity has to be done,  ie, do they inhibit Na pump, PM Ca2+ pump, NCX, SOCC channels etc., using direct measurement. The findings that selective inhibitors did not potentiate LEN in killing resistant PTC cells did not add too much merit (Fig3BC) if the above direct assays are not performed.

 > Reply: Thank you for your professional comment. I totaly agree with your opinion. We are not focussed enzyme activity measurement assy. Just our intention, present study showed that enhancement of SERCA expression level was pivotal events under lenvatinib treated conditions in patient-derived lenvatinib-resistant PTC cells. The study findings significantly contribute to identifying SERCA as a critical regulator of cytosolic free calcium-mediated apoptosis not calcium channel, exchanger and PMCA under acute endoplasmic reticulum stress conditions, such as anti-cancer drug, lenvatinib treatment. Current study shows SERCA could be a pivotal regulator in lenvatinib resistant PTC under acute ER stress conditions. In this study used L-type calcium channel blocker (CCB; bepridil, verapamil, or nifedipine), NCX (Na+/Ca2+ exchanger) inhibitor (KB-R7943), PMCA inhibitor (Caloxin2a1), PMCA inhibitor was no significantly infuenced to regulation overburdened cytosolic free calcium under acute ER stress by lenvatinib treament.  

3) Fig.2and 3 B,C extremely difficult to read because of the small sizes of symbols.

> Reply: Thank you for your comment. Figure 2, Figure B and C was modified according to your professional opinion.

Importantly, I suppose the x-axis is the concentration of the novel compounds, then how about the LEN alone group (LEN also 0 - 200 uM?)?

> Reply: I am sorry for the unnecessary confusion caused. x-axis is means a dose of either agent alone or combination with lenvatinib. For example, in Figure 2B left, lenvatinib alone treatmetn group was showed dose-dependent manner 0-200 uM. L+C40, combination treatment with following lenvatinib and C40 treatment was respectively 0-200 uM.

Then, what was the LEN concentration when added together with novel compounds? What were the concentrations of other blockers or inhibitors, such as TSG and nifed etc?

> Reply: Lenvatinib treatment alone or combination treatment with calcium channel blocker or NCX inhibitor or PMCA inhibotor was respectively 0-200 uM.

4) I find the concentrations of drugs are missing in the figures or figure legends. This is an important issue.

> Reply: Thank you for your professional comment. We totaly agree with your opinion. according to your expert opinion, we added table for IC50 (Table 2 ).

5) Why are L-type channel blockers used? I wonder if PTC cells have L-type channels. Why not try SOCC or TRP blockers

> Reply: According to priviously published articles, in a significant number of cancer cells, CCB (calcium channel blocker) was used for leading of anti-cancer effect. Current study want to shows that NCX and PMCA inhibitors-included CCB were no significantly influenced to anti-cancer resistnat cancer cell under acute ER stress condition by treatment of lenvatinib. Consequently we excluded SOCC or TRP blockers. Your expert comment will perform to another project. Thank you again for your professional comment.

6) Western blot results should be quantitated. 

> Reply: According to your expert opinion, I have made corrections whole figures in current article.

Reviewer 2 Report

Comments and Suggestions for Authors

I have attached the comments. 

Comments on the Quality of English Language

The English of the manuscript needs to be edited. Some sentences are vague.

Author Response

Reviewer 2

In this study, the authors aimed to determine the signaling pathways activated in lenvatinib-sensitive and -resistant papillary thyroid cancer (PTC) cell lines. The RNA-seq analysis revealed higher expression of SERCA isoforms in resistant cells compared to sensitive cells. They designed two SERCA inhibitors and assessed their effects on cancer cell survival and downstream signaling pathways. They showed that, while SERCA inhibitors alone did not affect cancer cell viability, the combination of these inhibitors and Lenvatinib significantly reduced the viability of Lenvatinib-resistant cells. They also showed that this combination therapy increased the expression of an ER stress marker, CHOP, in Lenvatinib-resistant PTC cells. Finally, the in vivo experiments confirmed the in vitro findings, demonstrating the effects of the combination of SERCA inhibitors and Lenvatinib on reducing tumor volume and weight. Despite these interesting findings, several concerns that need to be addressed:

 > Reply: I don't know how to thank you enough for reviewing our manuscript. A point-by-point response to your expert comments was indicated below. Thank you again for your review. I hope you are always healthy and happy!!

Keywords:

  1. Replace “anticancer drug-resistant papillary thyroid cancer” with shorter term like “drug resistance”.

> Reply: I have made corrections according to your expert opinion.

Introduction:

  1. Add more information about the molecular mechanisms of drug resistance in PTC, clinical data on the rate of drug resistance, survival rates, etc., of PTC patients. The authors should emphasize the importance of their study.

> Reply: Line 57-65, according to your expert opinion, several sentences were added.

  1. In line 61, the authors stated, “This study aimed to identify small-molecule inhibitors to enhance treatment efficacy in lenvatinib-resistant PTC”. Before explaining the aim of the study, the authors should clarify the treatment strategies for PTC patients. They should explain the role of Lenvatinib in PTC treatment and discuss possible resistance mechanisms to Lenvatinib.

> Reply: Line 68-73, according to your expert opinion, several sentences were added.

Result 2.1: Patient-derived PTC cell lines and their properties

  1. The authors should specify their criteria for categorizing PTC patients as sensitive or resistant to Lenvatinib.

> Reply: Line 93-98, according to your expert opinion, several sentences were added.

  1. In line 73, the authors stated, “Three classes of PTC cell lines—YUMC-S-P2 (a patient-derived lenvatinib-sensitive PTC cell line), and YUMC-R-P7 and YUMC-R-P8 (lenvatinib-resistant PTC cell lines)—used in this study were derived from resected specimens…”. How did the authors characterize the isolated cells? They should include their characterization results in the manuscript.

> Reply: Line 93-98, I have made corrections according to your expert opinion.

  1. Table 1 lacks a caption or title.

> Reply: I am sorry for the unnecessary confusion caused by my mistake. I have made corrections according to your expert opinion.

Result 2.2: Distinctions in genetic alterations and activated signaling pathways between patient derived lenvatinib-sensitive and -resistant PTC cell lines

  1. This section can be divided into three parts: 1) genes related to cancer stemness, 2) genes regulating the EMT, and 3) genes involved in FGF/FGFR signaling pathways. The results should be re-written, first explaining each part and its importance in cancer biology and PTC, then presenting the results.

> Reply: Line 118-124, Thank you for your comment. The sentence was added according to your professional opinion.

  1. In line 96, the “5envatinib-resistant PTC” should be corrected to “lenvatinib-resistant PTC”.

> Reply: I have made corrections according to your expert opinion.

  1. In line 115, “calcium” is repeated twice.

> Reply: Thank you for your comment. I have made corrections.

  1. In line 117, the authors stated, “We hypothesize that these highly activated calcium-related genes and signaling pathways in lenvatinib-resistant PTC cells are crucial in enabling PTC cells to evade cytoplasmic calcium-mediated apoptosis under severe ER stress conditions induced by anti-cancer drug treatment such as Lenvatinib”. The authors should clarify and discuss the role of SERCA in calcium signaling, homeostasis, and calcium-mediated apoptosis.

> Reply: Line 151-157, Thank you for your comment. The sentence was added according to your professional opinion.

  1. In line 128, the authors stated, “The current results showes that the regulation of SERCA expression in lenvatinib-resistant PTC cells was one of the critical elements associated in prolonging survival under lenvatinib-treated conditions.”. The RNA-seq results only show higher expression of SERCA in resistant cells, and the authors did not include any data showing the critical role of SERCA in prolonging the survival of lenvatinib-resistant PTC cells in this section, Therefore, this statement should be re-written based on the results obtained in this section.

> Reply: I totaly agree with your comment. According to your expert opinion, the sentence was modified. Line 165-167,

Result 2.3: Identification of therapeutic molecules, candidates 40 and 42, based on SERCA structure through in-silico screening for suppression of lenvatinib-resistant PTC

  1. In line 137, the authors stated, “Notably, candidates 40 and 42 were identified and selected owing to their relatively high binding affinity to the molecular structure of SERCA”. The in-silico analysis results should be included in the manuscript.

> Reply: Line 174-186, Thank you for your comment. The sentence was added according to your professional opinion.

  1. In line 139, the authors stated, “These candidates showed considerable suppression of SERCA activity, leading to their selection as SERCA inhibitors in this study (Figure 3A)”. How did the authors measure SERCA activity? Figure 3A only shows the expression, not the activity of the proteins.

> Reply: Thank you for your comment. I totaly agree with your comment. Figure 2A was just showed expression level not activity. According to your expert opinion, the sentence was modified, line 188.

  1. In lines 142-145, the figure numbers are cited incorrectly (Figure 2A-C).

> Reply: Thank you for your comment. I have made corrections.

  1. In line 184, “thapsigargin” should be explained.

> Reply: Thank you for your comment. According to your expert opinion, sentence was added

  1. In line 188, the authors stated, “Unlike lenvatinib-sensitive PTC, lenvatinib-resistant PTC cells showed a marked increase in the expression of BCL-2 and SERCA1 among the SERCA isoforms when treated with lenvatinib.”.

1) The figure number should be cited,

> Reply: I have made corrections.

2) the intensity of the bands should be measured and compared with the other groups,

> Reply: Thank you for your comment. I have made corrections.

3) the authors did not include the control group (untreated) in this figure. If they want to conclude that Lenvatinib treatment causes this elevation, they should present the results of the untreated group for comparison.

> Reply: Thank you for your comment. According to your expert opinion, Figure 2C was modyfied.

  1. In line 190, the authors stated, “However, combination therapy with lenvatinib and the novel SERCA inhibitors (candidates 40 and 42) significantly increased markers of ER stress (CHOP) and apoptosis (cleaved-caspase 3) through the functional inhibition of SERCA (Figure 2C).”. According to the WB results, I do not see any increase in CHOP and cleaved caspase 3 expression. Moreover, cleaved caspase 3 levels should be compared to total caspase 3 levels.

> Reply: According to your expert opinion, Figure 2C was modyfied. Change of SERCA1, CHOP and cleaved-caspase 3 was showed by immunoblot in lenvatinib-sensitive or -resistant PTC under with or without lenvatinib. Cleaved-caspase 3 (17 kDa) figure was modyfied.

  1. In figure 2B, the lines are too close together and the groups are hardly distinguishable. I recommend using different colors to separate the groups.

> Reply: Thank you for your comment. According to your expert opinion, Figure 2B was modyfied.

  1. The caption of figure 2 is repeated twice.

> Reply: Thank you for your comment. I have made corrections.

Result 2.4: SERCA1 as a key player in lenvatinib-resistant PTC cells for prolonging survival under lenvatinib treatment

  1. In line 206, what does “PMCA” stand for?

> Reply: The present study first showed that enhancement of SERCA expression level was pivotal events under lenvatinib treated conditions in patient-derived lenvatinib-resistant PTC cells. The study findings significantly contribute to identifying SERCA as a critical regulator of cytosolic free calcium-mediated apoptosis not calcium channel, exchanger and PMCA under acute endoplasmic reticulum stress conditions, such as anti-cancer drug, lenvatinib treatment. In lenvatinib resistant PTC, “PMCA” means PMCA inhibitor (Caloxin2a1), PMCA inhibitor was no significantly infuenced to regulation overburdened cytosolic free calcium under acute ER stress

  1. In figure 3, the intensity of the bands should be measured and compared with the other groups.

> Reply: Thank you for your comment. I have made corrections.

  1. In figure 3D, the “Bepridil” group is not explained in the manuscript.

> Reply: Line 306-310, Thank you for your comment. I have added sentence about explain of “bepridil” treated group.

  1. In figure 3D, “YUMC-R-7” should be corrected.

> Reply: Thank you for your comment. I have made corrections.

Result 2.5: Targeted therapy in vivo treatment with novel candidates 40 and 42 in a patient-derived lenvatinib-resistant PTC cell mouse xenograft model

  1. In line 226, the authors stated, “These cells were administered lenvatinib either alone or in combination with SERCA inhibitors (thapsigargin as a positive control, along with candidates 40 and 42).”. The sentence does not make sense and should be corrected.

> Reply: Line 333-336, Thank you for your comment. I have made corrections.

  1. In lines 228-240, because the results for tumor volume and weight are the same, they should be written together.

> Reply: Line 346-349, Thank you for your comment. I have made corrections

  1. In figure 4, the lines are too close together and the groups are hardly distinguishable. I recommend using different colors to separate the groups.

> Reply: According to your expert opinion, whole figure 4 was modyfied.

Discussion:

The discussion is poorly written. More information should be added to support the results, and the results should be discussed clearly and in detail.

> Reply: Thank you for your comment. According to your expert opinion, sentence was added

Reviewer 3 Report

Comments and Suggestions for Authors

The authors aimed to identify small-molecule inhibitors to enhance treatment efficacy in Lenvatinib-resistant papillary thyroid cancer (PTC). The work by Kim and colleagues would be quite relevant in the context of the identification of new therapeutic strategies to fight resistance to anticancer drugs in TC. However, the text is a bit confused in some parts and the conclusions are not always supported by properly performed experiments.

Specific Major points

1.     I was unable to determine the concentration at which the drugs were used; this information should be included in the Materials and Methods section and in general in  all the manuscript.

2.     Figure 1B: It appears from the figure that the Twist 1 gene is upregulated in the resistant cell lines, but this data does not match whit what we observe in the heat map, especially when compared to other genes in the same chart, such as Zeb 1, Zeb 2, and the snail genes. Is there an error?

3.     Lines 114-117: The authors state that the various signaling pathways are more activated in the resistant cell lines compared to the sensitive ones; however, this data is not presented. Only the data for the resistant cell lines is provided. These data should be included.

4.     The molecular weights should be included alongside the proteins in all the Western blot.

5.     In Figure 2B, and more generally in all the graphs where curves are presented, the curves are not easily distinguishable and should be made more discernible, perhaps by changing the colors of the curves and making the legend and experimental data points more prominent.

6.     Figure 2C: The authors state in the text that the combination of lenvatinib and SERCA inhibitors increases the levels of CHOP and cleaved caspase-3. However, this conclusion cannot be drawn without a control, i.e., untreated cells, to determine the baseline levels of these two proteins.

7.     Figure 3A: the quality of the images is not suffient to allow to drive conclusion. Add quantification graphs.

Minor points

Results

Line 86: “Anti-cancer drug-resistant cancer cells” is redundant.

Line 96: change “top” with “bottom”; there is a mistake “5envatinib”

Lines 99-100: “and additional EMT markers (SNAIL1, SNAIL2, ZEB1, ZEB2, and TWIST1)” is a repetition of lines 94-96 .

Line 100:  change “bottom” with “top”.

Line 140: the authors said “Figure 3A” but it should be “Figure 2A”, consequently you have to remove “Figure 2A” from line 143.

Lines 155-164: It’s a repetition of lines 147-155, remove them.

Line 218: correct "drug-resistant"

Comments on the Quality of English Language

The quality of the English language used is good.

Author Response

Reviewer 3

The authors aimed to identify small-molecule inhibitors to enhance treatment efficacy in Lenvatinib-resistant papillary thyroid cancer (PTC). The work by Kim and colleagues would be quite relevant in the context of the identification of new therapeutic strategies to fight resistance to anticancer drugs in TC. However, the text is a bit confused in some parts and the conclusions are not always supported by properly performed experiments.

> Reply: I don't know how to thank you enough for reviewing our manuscript. A point-by-point response to your expert comments was indicated below. Thank you again for your review. I hope you are always healthy and happy!!

Specific Major points

I was unable to determine the concentration at which the drugs were used; this information should be included in the Materials and Methods section and in general in all the manuscript.

> Reply: Thank you for your professional comment. According to your expert opinion, we added table for IC50 (Table 2 ).

  1. Figure 1B: It appears from the figure that the Twist 1 gene is upregulated in the resistant cell lines, but this data does not match whit what we observe in the heat map, especially when compared to other genes in the same chart, such as Zeb 1, Zeb 2, and the snail genes. Is there an error?

> Reply: I am sorry for the unnecessary confusion caused by my mistake. I have made corrections according to your expert opinion in figure 1.

  1. Lines 114-117: The authors state that the various signaling pathways are more activated in the resistant cell lines compared to the sensitive ones; however, this data is not presented. Only the data for the resistant cell lines is provided. These data should be included.

> Reply: Thank you for your comment. Signaling pathways related to calcium and cancer stemness, including Notch, Wnt, PPAR, PI3K/Akt, and TGF/SMAD, were significantly more activated in lenvatinib-resistant PTC compare than in lenvatinib-sensitive PTC. Figure 1C was normalized to Notch, Wnt, PPAR, PI3K/Akt, and TGF/SMAD signaling pathways of lenvatinib-sensitive PTC.

  1. The molecular weights should be included alongside the proteins in all the Western blot.

> Reply: Thank you for your professional comment. I have made corrections whole figures in current article.

  1. In Figure 2B, and more generally in all the graphs where curves are presented, the curves are not easily distinguishable and should be made more discernible, perhaps by changing the colors of the curves and making the legend and experimental data points more prominent.

> Reply: According to your expert opinion, I have made corrections whole figures in current article.

  1. Figure 2C: The authors state in the text that the combination of lenvatinib and SERCA inhibitors increases the levels of CHOP and cleaved caspase-3. However, this conclusion cannot be drawn without a control, i.e., untreated cells, to determine the baseline levels of these two proteins.

> Reply: Thank you for your professional comment. According to your expert opinion, no exposure lenvatinib data was added in ‘Figure 2C’.

  1. Figure 3A: the quality of the images is not suffient to allow to drive conclusion. Add quantification graphs.

 > Reply: Thank you for your professional comment. I have made corrections in ‘Figure 3A’.

Minor points

Results

Line 86: “Anti-cancer drug-resistant cancer cells” is redundant.

Reply: I have made corrections according to your expert opinion.

Line 96: change “top” with “bottom”; there is a mistake “5envatinib”

Reply: I have made corrections according to your expert opinion.

Lines 99-100: “and additional EMT markers (SNAIL1, SNAIL2, ZEB1, ZEB2, and TWIST1)” is a repetition of lines 94-96 .

Reply: Thank you for your professional comment. I have made corrections according to your expert opinion.

Line 100:  change “bottom” with “top”.

Reply: According to your expert opinion, I have made corrections.

Line 140: the authors said “Figure 3A” but it should be “Figure 2A”, consequently you have to remove “Figure 2A” from line 143.

Reply: I have made corrections according to your expert opinion.

Lines 155-164: It’s a repetition of lines 147-155, remove them.

Reply: I have made corrections.

Line 218: correct "drug-resistant"

Reply: Thank you for your professional comment. According to your expert opinion, I have made corrections in whole current article.

Reviewer 4 Report

Comments and Suggestions for Authors

In this article, the authors have identified and studied the effects of two active compounds presented as SERCA inhibitors in combination with lenvatinib on papillary thyroid cancer models.

The introduction seems a little short, and could do more to emphasize the role of lenvatinib and targeted therapies in the treatment of papillary cancer (a general reminder of the management of these cancers, including the role of surgery, would be very welcome).

Also, in the second paragraph of the introduction, a slightly more substantial contextualization of the physiological role of SERCA, its possible deregulation and pathophysiological roles in cancers, would be appropriate.

- line 87: the sentence ending with “several studies” calls for associated bibliographical references.

- line 96: please write « In lenvatinib-resistant PTC ».

- line 141: candidates 40 and 42 are definitely not “novel therapeutic small molecules”: they are isoflurane and ethacrynic acid respectively: this remark highlights all the limitations of this work.

The major issue with this work is that it has been carried out without once taking into account the fact that the two candidates identified are known molecules, used in therapeutics to date (anesthetic and diuretic), and that they therefore have their own pharmacology, which would undeniably have an impact on their possible use in combination with lenvatinib.

Furthermore, absolutely no information is given on the in silico screening process: which molecule banks were used, with what criteria, and so on. It is imperative that the authors consider the chemistry aspects of this work, and not just the biology side, if it is to be published.

Important questions arising from these remarks also need to be answered before we can advance the efficacy of these two derivatives, starting with their intrinsic cytotoxicity on healthy cells at the concentration studied here. A major rework of the manuscript and additional experimentation should be carried out in this direction.

Comments on the Quality of English Language

Minor editing of English language required.

Author Response

Reviewer 4

In this article, the authors have identified and studied the effects of two active compounds presented as SERCA inhibitors in combination with lenvatinib on papillary thyroid cancer models.

 > Reply: I don't know how to thank you enough for reviewing our manuscript. A point-by-point response to your expert comments was indicated below. Thank you again for your review. I hope you are always healthy and happy!!

The introduction seems a little short, and could do more to emphasize the role of lenvatinib and targeted therapies in the treatment of papillary cancer (a general reminder of the management of these cancers, including the role of surgery, would be very welcome).

> Reply: Line 57-65, 68-73, according to your expert opinion, several sentences were added.

Also, in the second paragraph of the introduction, a slightly more substantial contextualization of the physiological role of SERCA, its possible deregulation and pathophysiological roles in cancers, would be appropriate.

> Reply: Line 75-78, according to your expert opinion, several sentences were added.

- line 87: the sentence ending with “several studies” calls for associated bibliographical references.

> Reply: Thank you for your comment. I have made corrections.

- line 96: please write « In lenvatinib-resistant PTC ».

> Reply: Thank you for your comment. I have made corrections.

- line 141: candidates 40 and 42 are definitely not “novel therapeutic small molecules”: they are isoflurane and ethacrynic acid respectively: this remark highlights all the limitations of this work.

> Reply: I totaly agree with your expert comment. ‘Novel’ was deleted in whole current article. Current study used word ‘novel’ was intended to new therapeutic approaches for refractory cancers. I am sorry for the unnecessary confusion caused by my reckless word selection.

The major issue with this work is that it has been carried out without once taking into account the fact that the two candidates identified are known molecules, used in therapeutics to date (anesthetic and diuretic), and that they therefore have their own pharmacology, which would undeniably have an impact on their possible use in combination with lenvatinib.

> Reply: I totaly agree with your expert comment. Current research wanted to show SERCA could be pivotal regulator in lenvatinib resistant papillary thyroid cancer cell. Candidate 40 and 42 (isoflurane and ethacrynic acid) was used for only functional inhibitors to SERCA. I absolutely agree with your expert opinion, I'll make sure to reflect it in the ongoing our research. SERCA is widely recognized as a key regulator of cytosolic free calcium under severe ER stress conditions. However, a cardiac dysfunction was inevitable in vivo because of non-specific inhibition of SERCA isoforms by conventional SERCA inhibitors. I will proceed with the ongoing research in consideration of your expert opinion (impact on their possible use in combination with lenvatinib).

Furthermore, absolutely no information is given on the in silico screening process: which molecule banks were used, with what criteria, and so on. It is imperative that the authors consider the chemistry aspects of this work, and not just the biology side, if it is to be published.

> Reply: Thank you for your comment. According to your expert opinion, sentence of in silico screening process and result was added in ‘4. Materials and Methods section, line 574-589; 4.13. Virtual screening with chemical binding similarity’ and ‘2. Results section, line 174-186; 2.3. Identification of therapeutic molecules, candidates 40 and 42, based on SERCA structure through in-silico screening for suppression of lenvatinib-resistant PTC,’ respectively.

Important questions arising from these remarks also need to be answered before we can advance the efficacy of these two derivatives, starting with their intrinsic cytotoxicity on healthy cells at the concentration studied here. A major rework of the manuscript and additional experimentation should be carried out in this direction.

> Reply: Thank you for your comment. According to your expert opinion, modified whole article and additional experimentation carried out according your expert direction. Combination treatment with levatinib and candidates of IC50 table was added. Moreover, cell viability assay in normal cell (three parathyroid cells) was carried out for screening of intrinsic cytotoxicity about candidate 40 and 42 alone treatment respectively. This figure and result were showed to supplementary figure1 and results section. Candidate 40 and 42 alone treatment respectively was no considerably influenced to normal parathyroid cell in a dose-dependent manner. Lenvatinib treatment alone was showed siginificantly suppressed to cell viability of normal parathyroid cell (please see below supplementary figure 1 ). We carried out screening of intrinsic cytotoxicity of C40 and C42 in normal cell (three patient-derived parathyroid cell). C40 and C42 alone treatment respectively was no considerable influenced to three parathyroid cells. These results were added supplementary figure in current article.

Round 2

Reviewer 2 Report

Comments and Suggestions for Authors

The comments have been attached 

Comments on the Quality of English Language

The English of the paper should be improved and edited.

Author Response

Reviewer 2

 The authors have made some improvements to the manuscript, but I believe it still lacks sufficient background information. Additionally, several of my previous comments have not been adequately addressed. Below are the specific comments that need further revision, as the provided responses are unsatisfactory:

  • Clarification of treatment strategies and molecular mechanism of drug resistance in PTC: “In line 61, the authors stated, “This study aimed to identify small-molecule inhibitors to enhance treatment efficacy in lenvatinib-resistant PTC”. Before explaining the aim of the study, the authors should clarify the treatment strategies for PTC patients. They should explain the role of Lenvatinib in PTC treatment and discuss possible resistance mechanisms to Lenvatinib”

> Reply: In ‘ijms-3100359 R2’, line 57-65, 68-73, According to your expert opinion, several sentences were added.

  • Role of SERCA in calcium signaling: “In line 117, the authors stated, “We hypothesize that these highly activated calcium-related genes and signaling pathways in lenvatinib-resistant PTC cells are crucial in enabling PTC cells to evade cytoplasmic calcium-mediated apoptosis under severe ER stress conditions induced by anti-cancer drug treatment such as Lenvatinib”. The authors should clarify and discuss the role of SERCA in calcium signaling, homeostasis, and calcium-mediated apoptosis.”

> Reply: In ‘ijms-3100359 R2’, line 155-160, According to your expert opinion, several sentences were added.

  • “The discussion is poorly written. More information should be added to support the results, and the results should be discussed clearly and in detail.”

> Reply: Thank you for your comment. According to your expert opinion, sentence was added

In line 73, the authors stated, “Three classes of PTC cell lines—YUMC-S-P2 (a patient-derived lenvatinib-sensitive PTC cell line), and YUMC-R-P7 and YUMC-R-P8 (lenvatinib-resistant PTC cell lines)—used in this study were derived from resected specimens…”. How did the authors characterize the isolated cells? They should include their characterization results in the manuscript.

Given the heterogeneous nature of tumor cell populations, the authors should clarify which markers were used to identify the specific cells they intended to work with.

> Reply: In ‘ijms-3100359 R2’, line 93-102, According to your expert opinion, sentence and supplementary figure 2 were added in main article and supplementary informations.

In line 139, the authors stated, “These candidates showed considerable suppression of SERCA activity, leading to their selection as SERCA inhibitors in this study (Figure 3A)”. How did the authors measure SERCA activity? Figure 3A only shows the expression, not the activity of the proteins.

The term 'expression' would be more appropriate than 'function'

> Reply: In ‘ijms-3100359 R2’, line 192, I have made corrections according to your expert opinion.

In line 206, what does “PMCA” stand for?

The full name of PMCA should be provided.

> Reply: In ‘ijms-3100359 R2’, line 300, I have made corrections according to your expert opinion.

In lines 228-240, because the results for tumor volume and weight are the same, they should be written together.

This comment has not been addressed

> Reply: I have made corrections according to your expert opinion.

Reviewer 3 Report

Comments and Suggestions for Authors

The authors have promptly made the requested changes. Congrats!

Author Response

Reviewer 3

The authors have promptly made the requested changes. Congrats!

 > Reply: I don't know how to thank you enough for reviewing our manuscript. Thank you again for your expert comments. I hope you are always healthy and happy!!

Reviewer 4 Report

Comments and Suggestions for Authors

I take note of the changes made by the authors. Nevertheless, the names of the two identified candidates 40 and 42 are still not given anywhere in the manuscript (isoflurane and ethacrynic acid, respectively). It is essential to mention their names in the manuscript.

Author Response

Reviewer 4

I take note of the changes made by the authors. Nevertheless, the names of the two identified candidates 40 and 42 are still not given anywhere in the manuscript (isoflurane and ethacrynic acid, respectively). It is essential to mention their names in the manuscript.

> Reply: Line 34, Thank you for your comment. According to your expert opinion, the names of the two identified candidates 40 and 42 was added in main article.
